# Electrical Impedance Tomography Technical Contributions for Detection and 3D Geometric Localization of Breast Tumors: A Systematic Review

**DOI:** 10.3390/mi13040496

**Published:** 2022-03-23

**Authors:** Juan Carlos Gómez-Cortés, José Javier Díaz-Carmona, José Alfredo Padilla-Medina, Alejandro Espinosa Calderon, Alejandro Israel Barranco Gutiérrez, Marcos Gutiérrez-López, Juan Prado-Olivarez

**Affiliations:** Departamento de Ingeniería Eléctrica y Electrónica, Tecnológico Nacional de México en Celaya, Celaya 38010, Mexico; m1703042@itcelaya.edu.mx (J.C.G.-C.); javier.diaz@itcelaya.edu.mx (J.J.D.-C.); alfredo.padilla@itcelaya.edu.mx (J.A.P.-M.); alejandro.espinosa@crodecelaya.edu.mx (A.E.C.); israel.barranco@itcelaya.edu.mx (A.I.B.G.); marcos_22_11_88@hotmail.com (M.G.-L.)

**Keywords:** impedance tomography, breast cancer, breast tumor, geometric localization, cancer detection

## Abstract

Impedance measuring acquisition systems focused on breast tumor detection, as well as image processing techniques for 3D imaging, are reviewed in this paper in order to define potential opportunity areas for future research. The description of reported works using electrical impedance tomography (EIT)-based techniques and methodologies for 3D bioimpedance imaging of breast tissues with tumors is presented. The review is based on searching and analyzing related works reported in the most important research databases and is structured according to the Preferred Reporting Items for Systematic Reviews and Meta-analysis (PRISMA) parameters and statements. Nineteen papers reporting breast tumor detection and location using EIT were systematically selected and analyzed in this review. Clinical trials in the experimental stage did not produce results in most of analyzed proposals (about 80%), wherein statistical criteria comparison was not possible, such as specificity, sensitivity and predictive values. A 3D representation of bioimpedance is a potential tool for medical applications in malignant breast tumors detection being capable to estimate an ap-proximate the tumor volume and geometric location, in contrast with a tumor area computing capacity, but not the tumor extension depth, in a 2D representation.

## 1. Introduction

Even with current technological and scientific advances, cancer remains as one of the deadliest diseases worldwide [1]. Nearly eight million women around the world are diagnosed with some type of cancer. Breast cancer is the leading cause of death in women [2]. The highest incidence appears in women over 40 years old [3,4]. Timely detection of breast cancer notably helps to reduce the mortality rate [5,6].

An EIT (electrical impedance tomography) based screening test is an imaging technique that estimates the distribution of electrical conductivity in a body by measuring the actual impedance values through on-surface electrodes [7,8,9,10,11,12,13], which has been applied in breast tumor detection [14,15]. The EIT has gained interest in the medical field [16] due to its low cost, safety due to its operation without ionizing radiation, potential for portability, and miniaturization [17].

In vitro tissue characterization [18] has been performed to identify the difference between healthy and malignant tissues [19], and experimentation has shown measurement changes in tissues as the sampling frequency is varied [20,21,22,23]. There are significant differences in electrical impedance properties between a benign tumor and a malignant tumor [24,25,26,27,28,29,30,31]. The value of the tissue conductivity and impedance change as the cellular structure that conditions the electric current path is altered, EIT current paths, and equipotential surfaces are functions of the unknown resistivity distribution [32]. This results in a nonlinear reconstruction problem [33] where numerical algorithms [34] or finite element meshing [10] must be considered for resolution. Recently, efforts have been focused on using biotechnology in the medical field that improve diagnosis and facilitate the treatment of tumors [35,36,37,38].

An important feature when using EIT is image reconstruction. The 2D image representation systems start from the assumption that the current flow between electrodes is restricted to the image plane. This assumption is valid as an approximation for experimentation using shallow phantoms but is not realistic for a medical application in breast tumor detection. Ignoring current flow through the out-of-plane volume results in accuracy loss in the reconstructed images. A 3D image representation contains more information obtained from reconstructed image than a 2D one, but with an increasing complexity [7]. It is important to take into account the increased requirements in a 3D system such as a larger number of sensors, more computational processing workload, and the use of an adequate reconstruction technique. There are many reconstructing algorithms available, for instance the use of a dynamic imaging method allows a better accuracy by denoising the processed data [39].

Impedance measuring acquisition systems focused on breast tumor detection, as well as image processing techniques for 3D imaging are reviewed in this paper. The main EIT systems’ inconvenience is reconstructed images with low resolution, so techniques and methods to reduce the low-resolution effects are described to identify potential improvements for future EIT systems applied to breast tumors detection.

This paper describes a systematic review of EIT-based techniques and methods for 3D impedance imaging in breast with tumors. The goal is focused on presenting a main contributions analysis of EIT based detection and 3D geometric localization of breast tumors. This analysis provides a list of contributions made since 2015 as a complement to the review article published by Zain and Kanaga in 2015 [40]. This review is intended to identify potential opportunity areas for future research. The topics covered include technical characteristics (number of electrodes, type of electrode array, electrode current injection values and frequency), and scopes (minimum tumor detection size, representation method for measured values and clinical test results that the process should be achieving).

## 2. Materials and Methods

In order to review the state of the art in the detection and 3D geometric localization of breast tumors based on bioimpedance breast representation, two main objectives are defined in this work: (a) to describe the reported methodology and electronic instrumentation for breast tumor detection and geometric localization; and (b) to present an advantages/disadvantages analysis of reported related works, in order to identify potential research opportunity areas for new contributions.

### 2.1. Search Strategy

Several proposals have been made for the detection of breast tumors based on EIT. The review described in this paper is based on an exhaustive search and systematic analysis of scientific papers reporting procedures and results using EIT as a method of breast tumor depiction. The search of the analyzed reported works was performed in the main research databases (Science Direct, PubMed, EBSCO host, Web of Science, IEEE, Google Scholar, CORE, DOAJ and BASE).

### 2.2. Inclusion Criteria

The review covered works published since 2015 and its inclusion criteria considered: experimental studies, clinical trials and proposal simulations, studies using EIT as a breast tumor locating technique, and applied studies for breast tumor detection. Some articles were discarded due to not reporting enough information (e.g., technical characteristics of the proposed system, simulation, experimentation or clinical trials in the validation). The papers reporting EIT as a reference with other tomography techniques, (missing information about the technical capabilities compared with the EIT system were also discarded).

### 2.3. Methodological Quality

In order to ensure review integrity, transparency, quality, and consistency, the PRISMA statement standards were followed [41].

### 2.4. PRISMA Flowchart

As result of performing an intensive search in the main databases, a total of 218 EIT related papers were considered in the initial identification stage. The results in the searching stages are depicted in Figure 1 using the PRISMA format.

A total of 33 articles was discarded as duplicates. Then 118 articles were excluded since they were reported prior 2015. Finally, 48 articles were excluded since at least one validation procedure (simulation, experimental or clinical trials) was not depicted. The articles excluded in this stage were the ones not focused on the detection of human breast tumors or that did not reflect the technical characteristics of the described proposals (i.e., electrode arrangement, working frequency or value of the electric current injection).

## 3. Results

As a result, a total of 19 papers meeting the inclusion criteria were included in the review (Figure 1) which are presented in Table 1.

In medical applications, EIT as an imaging technique presents low reconstruction quality and requires higher computational resource cost [42,43,44,45] than other imaging techniques such as computed tomography (CT), magnetic resonance imaging (MRI), and nuclear magnetic resonance (NMR).

The EIT image reconstruction quality is determined by several parameters such as electric current injection value, measurement patterns of injected electric current and applied voltage, measurement accuracy, as well as the number and arrangement of measurement electrodes [46]. The main characteristics of the signal acquisition systems reported by the papers included in the review are depicted in Table 1, where not reported information is denoted by the symbol “-”. It is also shown if the reported systems are designed and built meeting medical standards as a reference for subsequent use in clinical trials.

**Table 1 micromachines-13-00496-t001:** Main electrical characteristics of EIT-based systems included in the review.

Author (Year)	Electrode Arrangement	Working Frequency	Electric Current Injection	Medical Standard Validation
Choridah et al. (2021) [47]	16 and 32 electrodes	-	-	-
Gomes et al. (2020) [42]	16 electrodes in a ring a array	-	-	-
Hu Jing et al. (2020) [48]	16 electrodes in a ring array	50 Hz to 250 kHz	1–7 mA	SwissTom Pioneer commercial system
Lee Jaehyuk et al. (2020) [49]	16 electrodes distributed in 2 levels	10 kHz to 10 MHz	0.1 to 3 mA pp	-
Mansouri et al. (2020) [50]	4 electrodes in a ring array	1 kHz	0.9 mA	Study approved by Research Ethics Committee in Health and Science Disciplines
Murillo-Ortiz et al. (2020) [51]	2 electrodes one on each arm	50 kHz	0.5 mA	MEIK v.5.6 commercial system
Chen et al. (2020) [52]	16 electrodes in a ring array	-	-	-
Gutierrez et al. (2019) [53]	8 electrodes in a ring array	500 Hz, 1 kHz, 5 kHz And 10 kHz	60 μA pp	IEC/TS 60479-1
Rao et al. (2019) [54]	16 electrodes in a ring array	100 Hz to 10 MHz	1.2 mA pp	-
Mothi et al. (2018) [55]	16 electrodes in a ring array	260 kHz	7 mA	SwissTom commercial system
Wu et al. (2018) [56]	16 microelectrodes in A ring array	10 kHz	-	-
Zarafshani et al. (2018) [57]	85 electrodes in a planar array	10 kHz to 3 MHz	10 mA	IEC 60601-1
Singh et al. (2017) [43]	16 electrodes in a ring array	1kHz to 1 MHz	0.5 mA pp	-
Yunjie Yang et al. (2016) [58]	16 microelectrodes in a ring array	10 kHz	1.5 mA pp	Class II, type BF
Murphy et al. (2016) [59]	16 electrodes in a ring arrangement distributed in 2 rings (rotary system)	10 kHz, 100 kHz, 1 MHz and 10 MHz	-	-
Hong Sunjoo et al. (2015) [60]	90 electrodes distributed in multiple levels using ring arrangement	100 Hz to 100 kHz	10 to 400 μA pp	IEC 60601-1
Khan Shadab et al. (2015) [61]	16 electrodes in a ring array	1 kHz to 100 kHz	100 μA rms	IEC 60601
Zhang et al. (2015) [62]	85 electrodes in a planar array	500 kHz	-	-
Halther et al. (2015) [63]	16 electrodes in a ring array	127 kHz	1 V pp	Institutional Review Board-approved study at Dartmouth-Hitchcock Medical Center (Lebanon, NH, USA).

The following table shows the technical scope of the proposals analyzed in order to compare the detection capacity (minimal tumor size) of each system. Additionally, the proposal validation of the systems is presented: simulation (S), experimental (E), or clinical trial (CT). The imaging technique, the proposal validation and the tumor size for each of the reviewed papers are presented in Table 2. A 2D reconstruction is usually referred to as a reconstruction of a single layer (plane) of the total breast volume, a 3D reconstruction is referred to as a total breast reconstruction or the use of multiple layers of different heights of the breast.

Some of the proposed systems were used in clinical trials to measure their capacity. Percentage value of the sensitivity and specificity of the reviewed works, which applied the EIT in clinical tests in the detection of breast tumors are presented in Table 3.

## 4. Discussion

Approximately 81% of the reviewed papers use in the impedance measurement an electrodes ring arrangement, which distribution allows to be properly positioned on the anatomy of the studied breast. Specifically, a 16-electrode ring array is predominantly employed (81% of the analyzed works). The greater number of electrodes, the higher resolution for tumor representation (smaller-tumors detection), but the higher computational resources and processing time. In average, a working frequency value within kHz range is observed for most of the studied works. The smallest tumor size reported in the reviewed papers is 5 mm. A comparison between an EIT system and an ultrasonic one is presented by Choridah et al. (2021) [47], where technical features such as injection values and minimum detection size were omitted, in order to show the EIT imaging disadvantages (low resolution resulting in the absence of sharp edges in the image), the human tissue is emulated by using chicken tissue. It is recommended to extend the scope of this study in order to compare the minimum sizes of tumors detectable by both techniques (electrical impedance tomography and ultrasonography) to determine the EIT capacity in clinical diagnosis.

Image processing based on neural networks to reduce the impact of the EIT characteristic low resolution is proposed in Gomes et al. (2020) [42], which is a work to be considered for diagnostic improvement in medical applications. An experimental stage or clinical trial results would be recommended in order to analyze the effect on the image reconstruction quality and detection size improvement compared with artificial intelligence networks. The difference between using a ring electrode distribution and a deformable electrode distribution that showing an improvement in the image reconstruction quality due to the mesh variety is explained in Hu Jing et al. (2020) [48]. A minimum detectable tumor size comparison between the traditional EIT technique and the use of “boundary shapes in deformable EIT” is suggested. This test would allow to quantitatively establish the minimum detectable tumor size with the proposed technique.

The use of additional electronic instrumentation to an EIT system (i.e., wide dynamic range LNA (WDR-LNA), dual mode driver (DM-Driver), and phase compensation loop (PCL)) for improving the image accuracy by minimizing stray capacitance is proposed by Lee Jaehyuk et al. (2020) [49]. This proposal achieves an accurate breast cancer detection system but at the expense of greater complexity. It would be useful to define the required hardware aspects of this proposal such as the system size and the proposed patient mounting to establish the system’s portability.

The impedance difference between the two breasts to determine the presence of a tumor is used in Mansouri et al. (2020) [50], this additional parameter may improve the initial diagnosis. A further study is proposed to determine if the bioimpedance difference between the two breasts is correlated to the tumor size. If this correlation is confirmed the application of such proposal to new EIT systems, it would be possible to corroborate the detected tumor size. In addition, it is recommended to present the sensitivity and specificity parameters for clinical test validation.

The use of a field programmable gate arrays (FPGA)-based processing system can help to reduce the analysis computing time compared with a sequential processing-based system [50,54,61]. A post-detection breast imaging reporting and data system (BI-RADS) classification is proposed by Murillo-Ortiz et al. (2020) [51], which is based on a tumor physical evaluation such as shape, contour, surrounding tissues integrity and electrical properties, among others. Once the tumor is classified, a diagnosis procedure can be suggested to the patient, such as a mammography or a biopsy procedure, in order to prescribe the right treatment for a benign or malign tumor. The use of deep learning techniques is proposed in Chen et al. (2020) [52] due to its fast inferences on object detection, image segmentation and classification. This system explicitly recognizes multiple cells in an unseen micro scale. The proposal has potential application in other systems to improve diagnosis by detecting tumors that are not visible in initial imaging.

The recommendation of using normalized impedance plots by Gutierrez et al. (2019) [53] allows the preservation of most information obtained from each measurement by the electrodes ring arrangement. It is suggested to present a 2D imagen reconstruction to validate the complementary information given by the normalized impedance plots in the breast reconstruction, and how this information can improve the diagnosis when it is applied to clinical trials.

The hyperparameters use to obtain the best reconstructed image is proposed by Mothi et al. (2018) [55], where commercial system is used for the measurements acquisition with the incorporation of a free software in the images generation. A pioneering system for real-time monitoring of a chemically insulted tumor cell, which was observed not to change its conductivity after chemical insult, is reported in Wu et al. (2018) [56]. This system sets a precedent for the development of smaller detection systems capable to real-time monitoring of chemical treatments. The system based on a medical standard for future clinical trials described by Zarafshani et al. (2018) [57] exposes the remarkable system feature of using a realistic electronic phantom (E-phantom) for validation experiments development. The E-phantom includes a complex circuit emulating the extra-cellular resistance, the intracellular tissue resistance and the membrane capacitance properties such as a real breast. It is proposed to present an agar breast phantom model with similar characteristics to the E-phantom realistic model in order to compare the obtained reconstructions. The proposal designed following the norms of a clinical standard so it can reach clinical trial tests.

The modular system reported by Singh et al. (2017) [43] can increase the number of electrodes and is a portable system that can be easily used in the medical field since it does not require specialized knowledge. The main strength of this proposal is that its architecture principal characteristics are low cost, continuous improvement, and ease of programming, which is based on the Raspberry pi board. The system presented by Yunjie Yang (2016) [58] allows detection of a tumor size down to 0.55 mm, the proposed EIT-based system was originally designed for cell representation with diameter of 15 mm and height of 10 mm, this size limitation should be considered when performing clinically live testing on patients. Murphy et al. (2016) [59] and Zhang et al. (2015) [62] explain the advantage of a rotary EIT system improving image contrast and an improved capability to distinguish between closely spaced inclusions (tumors) compared to traditional EIT. It is an alternative to achieve image quality improvement without increasing the number of electrodes as would be the case with a traditional EIT system.

A portable solution possibility for an EIT system is described in Hong Sunjoo (2015) [60] by using a proposed application to display the EIT analysis results in a mobile device. The recommendation is to perform clinical tests to research the patients’ reaction using this system and to evaluate potential disadvantages in the tests such as poor electrodes contact, tingling in patients, the need to use a conductive gel to perform tests, if the image reconstruction software is compatible with all mobile devices, etc.

The modular system proposed by Khan Shadab et al. (2015) [61], which allows upgradability and reconfigurability to avoid obsolescence. The use of digital filter in this system is proposed to reduce measurement noise. An explanation of the need to improve sensitivity and specificity of detecting malignant formations within the breast for EIT implementation in medical applications is given in Halther et al. (2015) [63].

Approximately 15% of the analyzed proposals reported 3D imaging (multiple layer or 3D reconstruction). The need to using a greater number of electrodes is due to increase the number of measurements and perform measurements in more breast regions. The reviewed works reporting a clinical trial are experimentally tested following a clinical safety standard, which allows the proposed system functionality validation with patients. Most of the EIT proposals included in this review do not lead to clinical trials, this fact makes difficult a results comparison with current and most used screening techniques, such as mammography, ultrasound and MRI [47,64,65,66,67,68]. MRI and breast ultrasound screening methods are able to detect tumors that are palpable, the MRI method has also demonstrated the capability to detect non-palpable tumors (detecting tumors as small as a 7.5 mm). To improve detection capability, breast ultrasound and MRI are combined with mammography [66]. Most of the systems reviewed in this paper are capable to detect a tumor size from 10 mm, this aspect should be considered when a comparison with other screening methods is carried out. Due to the clinical trials missing, or indeed demonstration of EIT effectiveness in clinical trials, it is not possible to make an evaluation based on statistical criteria of the reported systems, for instance features such as specificity, sensitivity, and predictive values.

## 5. Conclusions

The EIT-based systems seek to have an optimal number of electrodes that present a resolution that allows the analysis of a detected tumor. The technical characteristics of the different analyzed proposals have been detailed to provide a review of the technological achievements in the last six years. Most of the EIT systems use a 16-electrode standard in a ring array for breast measurement. In addition, a frequency operation in the kHz range is employed in the most of the systems and can operate at multiple frequencies. Clinical trials are required to consider statistical parameters in the comparison of the proposed systems. Only 20% of the reviewed articles concluded in clinical trials, this limitation does not allow comparative studies with other breast tumor detection methods. The future trend is to increase the EIT systems resolution in order to make reliable diagnostic tests, preserving the characteristics of portability and low cost. Multilevel electrode systems allow the geometric 3D localization of breast tumors, which is a research area of opportunity for diagnosis quality improvement. The 3D image reconstruction is not fully reported in the reviewed papers and it is normally limited to the representation of the plane covered by the electrode arrangement. One of the limitations to implement a multilevel system is the number of electrodes needed in addition to the increased computational resources required for the measurement and image reconstruction.

An EIT system design for detecting and locating breast tumors should consider parameters such as the meeting of a medical standard, system validation by clinical tests to obtain statistical parameters (i.e., sensitivity and specificity), and performance comparison with other breast tumor detection techniques. Another design consideration is obsolescence; a modular system provides upgradability and reconfigurability at the expense of portability and cost. Although processing speed is an important feature, it seems to be more significant in systems with a considerable number of electrodes. The use of digital processing techniques and neural networks seems to be a trend for the improvement of EIT. Future researches should focus on reaching clinical trials in order to identify more areas of opportunity and needs for EIT in a real clinical environment. Due to the limitations of the EIT technique (low resolution, image reconstruction complexity), EIT is still considered as a complement to studies such as mammography. Improving the image quality and proving the detection of smaller sized tumors can validate EIT as a reliable complement to mammography in clinical diagnosis. Although other detection methods are more accurate, EIT imaging offers a low-cost solution without tissue radiation exposure.

## Figures and Tables

**Figure 1 micromachines-13-00496-f001:**
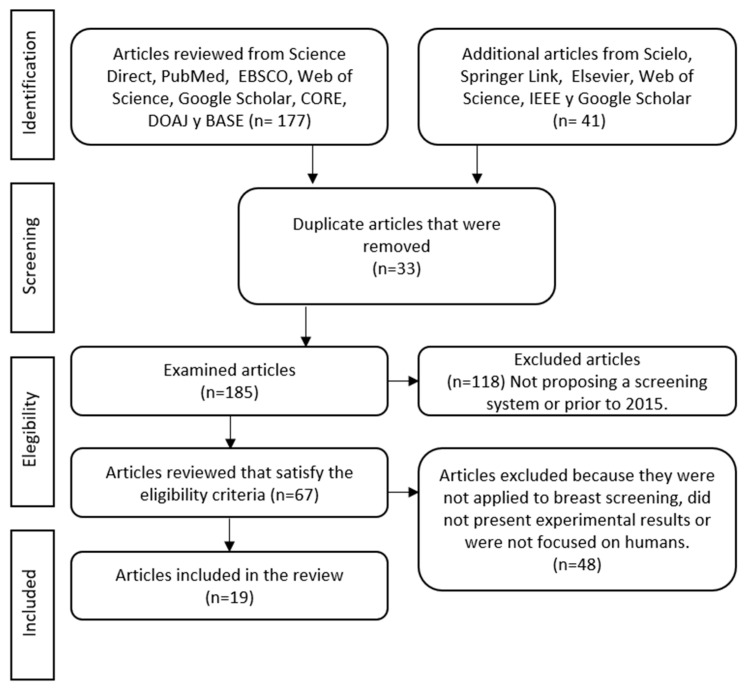
PRISMA flowchart for review paper selection.

**Table 2 micromachines-13-00496-t002:** Imaging technique, proposal, and tumor size detected applying EIT.

Author (Year)	Imaging Technique	Proposal Validation	Tumor Size
Choridah et al. (2021) [47]	Imaging using a single layer (2D)	E: Chicken phantom filled with an artificial solid tumor	-
Gomes et al. (2020) [42]	Imaging using a single layer (2D)	S: Images generated in MATLAB	-
Hu Jing et al. (2020) [48]	Imaging using a single layer (2D)	E: 3D printed samples and phantoms	From 5 mm.
Lee Jaehyuk et al. (2020) [49]	Imaging using a single layer (2D)	E: Agar phantom using carrots as tumors	From 5 mm.
Mansouri et al. (2020) [50]	Impedance measurements between left and right breast	CT: 40 women.	-
Murillo-Ortiz et al. (2020) [51]	Single layer imaging (2D), tumor classification	CT: 1200 women	-
Chen et al. (2020) [52]	Single layer imaging(2D) and image processing	E: Phantom in micro scale	-
Gutierrez et al. (2019) [53]	Normalized Impedance plots	E: Agar breast phantom model	From 10 mm.
Rao et al. (2019) [54]	Single layer imaging (2D)	E: Saline tank setup	From 13 mm.
Mothi et al. (2018) [55]	Single layer imaging on EIDORS Software (2D)	E: Gelatine breast phantom model	From 10 mm.
Wu et al. (2018) [56]	Single layer imaging (2D)	E: Miniature EIT sensor using solution	From 1.2 mm.
Zarafshani et al. (2018) [57]	Single layer imaging (2D)	E: E-phantom realistic model	-
Singh et al. (2017) [43]	Single layer imaging on EIDORS software (2D)	E: Plastic tank phantom and background solution	-
Yunjie Yang et al. (2016) [58]	Multiple layers imaging (3D)	E: Miniature phantom	From 0.55 mm.
Murphy et al. (2016) [59]	Imaging of electrical conductivity cross-section (2D)	E: Tank filled with saline solution	From 10 mm.
Hong Sunjoo et al. (2015) [60]	3D reconstruction on a mobile device (3D)	E: Agar breast phantom model	From 5 mm.
Khan Shadab et al. (2015) [61]	Single layer imaging (2D)	E: Tank filled with saline solution	From 25 mm.
Zhang et al. (2015) [62]	Multiple layers imaging (3D)	S: Digital breast phantom	From 15 mm.
Halther et al. (2015) [63]	Single layer imaging (2D)	CT: 19 women	From 20 mm.

**Table 3 micromachines-13-00496-t003:** Sensitivity and specificity percentage for reviewed papers using clinical trials.

Author (Year)	Sensitivity	Specificity
Mansouri et al. (2020) [50]	-	-
Murillo-Ortiz et al. (2020) [51]	85%	96%
Halther et al. (2015) [63]	77%	81%

## Data Availability

Not applicable.

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
