# Peer review of "Electrical Impedance Tomography Technical Contributions for Detection and 3D Geometric Localization of Breast Tumors: A Systematic Review"

_micromachines, 2022, doi:10.3390/mi13040496_

Round 1

Reviewer 1 Report

In this paper, the authors have made a review on “Electrical impedance tomography technical contributions for detection and 3D geometric localization of breast tumors”. Although much work has been done, I think there are two main problems to be solved.

1、This paper focuses on the review, i.e. the method and discussion. So, it is suggested to simplify the "introduction" and add more in-depth comments on literatures.

2、As a review article, I don't think the authors’comments are comprehensive, so I suggest increasing the number of literatures, especially the latest articles.

Author Response

Dear reviewer,

Thank you very much for your time and the very favorable comments on this work.

Reviewer 2 Report

This article presents the technical contribution of electrical impedance tomography in the detection and three-dimensional geometric localization of breast tumors.

The article is an interesting approach in terms of analyzing the medical research achievements in electrical impedance tomography. The effectiveness of this research is unfortunately limited in large part due to the lack of clinical trials, or indeed demonstration of its effectiveness in clinical trials.

Minor remarks:
1) I suggest describing in more detail on what basis and from what the scope of the selection of publications for the research problem analyzed. 
2) The authors could refer to other works showing the effectiveness of the presented method with other tomographic techniques.
3) The article needs minor language corrections.

Author Response

Dear reviewer.

Thank you very much for your positive comments on this review. We appreciate your comments and suggestions to improve the quality of this article.
